# Study of Phase Evolution Behavior of Ti6Al4V/Inconel 718 by Pulsed Laser Melting Deposition

**DOI:** 10.3390/ma16062437

**Published:** 2023-03-18

**Authors:** Yuanhao Wang, Xin Ye, Mingli Shi, Nanxu Pan, Peng Xia

**Affiliations:** 1School of Materials Science and Engineering, Shanghai University of Engineering Science, Shanghai 201620, China; wyh1998159357@163.com (Y.W.);; 2Shanghai Collaborative Innovation Center of Laser Advanced Manufacturing Technology, Shanghai 201620, China

**Keywords:** pulsed laser deposition, Ti6Al4V/Inconel 718 composition, phase evolution, microhardness

## Abstract

In this study, a pulsed laser was used as the heat source for the additive work. The Ti6Al4V/Inconel 718 alloy wire was deposited on the substrate by melting using a pulsed laser. Using the above method, single-layer and double-layer samples were printed. The sample material printed in this way is highly utilized. Compared to the complicated pre-preparation work of metal powder pre-mixing, this printing method is simple to prepare and only requires changing the wire feeding speed. The study of this paper provides a theoretical guide for the subsequent fusion deposition of heterogeneous wire materials. The samples were analyzed after molding using SEM, EDS and XRD to characterize the microstructure of the samples. The samples can be divided into three zones depending on the microstructure, the bottom columnar crystal zone, the middle mixed phase zone, and the bottom equiaxed crystal zone. From the bottom to the top of the sample, the phase microstructure changes as γ + Laves → α + β + Ti_2_Ni + TiNi + Ni_3_Ti → α + β. The hardness data show that the highest value in the transition zone is 951.4 HV. The hardness of the top part is second only to the transition zone due to a large number of equiaxed crystals. The bottom region is dominated by columnar crystals and is the softest of the three regions with the lowest hardness value of 701.4 HV.

## 1. Introduction

As one of the laser additive manufacturing technologies, laser melting deposition technology has a wide range of applications and can be targeted to produce parts for different working conditions. Muller et al. [1] have designed a part with adjustable alloy composition based on this technology. Such materials with a gradual change in composition are collectively referred to as functional gradient materials. The most remarkable technical feature of this material is the characterization along a certain direction of the object. The chemical composition or microstructure of the corresponding location within the material is gradually changed [2]. The functional gradient materials prepared by this technology can easily produce components with complex geometries to adapt to complex operating conditions. Traditional casting and other metallurgical techniques are not well suited to produce functional gradient materials for the increasingly complex commercial trade [3,4].

The current market for laser additive technology for the preparation of functional gradient materials is mainly used in the temperature difference between the two sides of the large or diverse working conditions of the parts. Among the typical application scenarios are gas turbines for aerospace propulsion, nuclear power plant insulation wall, and other complex environments [5,6].

In recent years, laser melting deposition technology has been widely used for the fabrication of different kinds of alloys. Shah et al. [7] successfully printed functional gradient materials in stainless steel and nickel alloys. Bobbio et al. [8] researched on Invar/Ti6Al4V, and Koikea et al. [9] researched Inconel625/316L. They all adopted laser melting deposition technology and successfully prepared functional gradient materials with different alloy compositions. Wei et al. [10]. prepared Inconel625/Ti6Al4V by laser preheating and laser deposition. However, the authors developed cracks at certain locations during subsequent processing when preparing unpreheated Inconel625/Ti6Al4V using the laser melting deposition technique. This is because laser melting deposition, as a rapid prototyping technique, has a very short heating to the cooling cycle. Parts made by this technique may have undesirable metal compounds within the microstructure and become unusable. Further studies by the above authors revealed that Mo and Cr elements have a significant influence on the formation of cracks in the samples. Shang et al. [11] found that the decrease in deformability of the β-phase of titanium alloys due to the large amount of Ni atoms in solid solution leads to a decrease in the mechanical properties of the samples. Sui et al. [12] found that Ni content in titanium alloys above 1.6 wt.% leads to the cracking of the samples. However, the above-mentioned researchers have studied less on the alloy part wan ith an equal volume ratio when studying gradient functional materials. Moreover, laser as an important heat input has not been explored much in the study. Therefore, this paper uses the pulsed laser deposition technique for additive manufacturing to investigate the effect of sample microstructure morphology on sample properties at medium ratio transition of functional gradient materials. Therefore, this paper uses the pulsed laser melting deposition technique for additive manufacturing by delivering the same volume ratio of metal material and printing it into the desired shape. The above samples will be used to study the histomorphology of samples from equal volume fractions of functional gradient materials.

In this study, the pulsed laser in laser additive manufacturing technology was used as the heat source. Unlike the conventional metal powder as the raw material, two different types of alloy filaments were used in this study for 3D printing and the feasibility of the technology was confirmed. The types of alloy wires were Inconel 718 alloy and Ti6Al4V alloy. This is due to the excellent corrosion resistance and good processing properties of Inconel 718 alloy. Additionally, high-temperature alloys are often used as transition materials in a high-temperature environment. This metal is the ideal experimental material for this study. Ti6Al4V alloy has a high specific strength and excellent processing properties and is commonly used as an engineered biomaterial [13,14]. When these two alloys are joined and fused by continuous laser, on the one hand, the coefficient of thermal expansion between the two metals is different. On the other hand, the intermetallic compounds increase during laser deposition and reduce the mechanical properties of the samples. Both of these aspects can cause a reduction in the overall performance of the sample or even failure. Therefore, compared to continuous laser, pulsed laser reduces the heat input to the workpiece, i.e., reduces the thermal stress. Thermal stress is an important parameter that affects cracking. In summary, this experiment using pulsed laser additive manufacturing of Inconel 718/Ti6Al4V is feasible on a theoretical basis.

## 2. Materials and Methods

### 2.1. Materials and Laser Processing

Figure 1 shows a picture of the experimental setup of the two-line parallel laser deposition system used in this study. Figure 1a shows the schematic diagram of the entire equipment of the printing platform, and Figure 1b shows the operation process of laser printing.

Unlike other studies where the raw material is metal powder, the metal wire can be processed effectively to avoid excessive losses. The material of the metal wire is dense and can be processed in such a way as to avoid as much air as possible from entering the sample and affecting the experimental results. Commercial Ti6Al4V and commercial Inconel 718 alloys with a diameter of 1.2 mm were supplied by VBC Group (Loughborough, UK). The chemical composition of the two alloy wires is shown in Table 1 and Table 2. Two alloy wires are fed into the molten pool at angles of 60° and 120° to the horizontal at the front and rear of the laser beam to align the wire tips with the center of the pool. Inconel 718 plate of dimension 200 mm × 120 mm × 6 mm was used as the substrate material. The distance between the wire nozzle and the substrate is kept at 8 mm to prevent the nozzle from being melted by the laser beam. All deposition processes are protected with high-purity argon gas. High-pressure argon gas is injected through a compression nozzle to protect the melt pool and the formed sample from the air.

In this study the average laser power was 480 W, the laser on time was 8 ms, and the laser waveform was a square wave with a frequency of 20 Hz. The overall laser system is a commercial product of the TRUMPF (Munich, GER), the product model is TruPulse556. The robotic arm is a commercial product from ABB (Zurich, CH), and the robot arm is set to move at a speed of 2.5 mm/s.

The experimental platform for automatic double filament deposition was built by integrating other equipment with the laser’s integrated system. The wire feeding speed was set to 60 cm/min in this experiment. The protective atmosphere was made of argon of 99.9% purity and the gas flow rate was set to 20/L.min^−1^. Two types of nozzles were set, the circular nozzle for adding the protective atmosphere to the melt pool and the square nozzle for protecting the formed sample. The above parameters were also collected in Table 3 for demonstration. The trajectory of the laser is the same as that of the robotic arm, the power of the pulsed laser is set to 480 W, and the shift speed is kept at 2.5 mm/s. The two wires can be fused to one piece better when printing is performed using the above parameters.

### 2.2. Wall Building

In this paper, two types of samples with an equal volume of Ti6Al4V/Inconel 718, single-layer, and double-layer, were printed by the intelligent platform built above.

Single-layer samples of Ti6Al4V/Inconel 718 of equal volume with a thickness of 4 mm and double-layer samples with a thickness of 8 mm were printed by the above platform. Figure 2 shows the sample diagram and the printed schematic. The average layer height of single-layer samples was 3.8 mm and the average layer height of double-layer samples was 7.8 mm as measured using vernier calipers. The lasers defocus on the substrate is set to +4 mm, expanding the range of the laser’s direct action on the substrate.

To fuse dissimilar metal materials and reduce cracking in conventional continuous lasers. Researchers usually need to preheat the substrate, the raw material. In contrast, the thermal physical parameters between titanium and nickel differ greatly, and the stresses generated in the bonding region of the two metals at the higher thermal input of the continuous laser pose greater difficulties for sample forming.

To prevent cracking when fusing dissimilar metals in conventional continuous laser. Researchers usually need to preheat the substrate and the raw material. As shown in Table 4 the thermal physical parameters between titanium and nickel metals differ significantly, and the stresses generated in the bonding region of the two metals pose greater difficulties for sample forming under the higher heat input of the continuous laser. The pulsed laser has a lower heat input under the same unit conditions and is now widely used for thin film fabrication and metal surface treatment [15]. Therefore, it is valuable to study the deposition of titanium–nickel alloy using a pulsed laser and to investigate the molding results.

To compensate for the error caused by the pulsed laser re-melting the first layer during the second layer of the sample, the printing height is compensated by reducing the laser lift height difference by 0.5 mm. Figure 2 shows the sample diagram and the printed schematic. The top sample is a single-layer sample, and the bottom is a double-layer sample. What was surprising was the appearance of a golden titanium oxide layer on the surface of the sample after it was formed. The oxide layer peeled off from the surface of the sample in the form of flakes. In additive manufacturing, such oxides can cause defects to appear within the sample microstructure, forming the origin of cracks. When printing samples in multiple layers, oxides can create additional difficulties for laser penetration, affecting the bond between layers [17].

### 2.3. Sample Characterization Pre-Treatment

After successful sample preparation, the sample is removed from the substrate pair. The removed samples were cut along the cross-section. The cut sections were polished with SiC to 2000 grit size. Further polishing with a suspension with a particle size of 1.5. Finally, the samples were etched with Kroll reagent (HF:HNO_3_:CH_3_CH_2_OH=1:2:7). The microstructure of the sample was analyzed by scanning electron microscope (SEM) and energy dispersive spectrometer (EDS). The microhardness of the composites was measured using an Hv-1000 Vickers microhardness tester, of which the loading force was 200 g and dwell time was 15 s. The phase composition of the sample was analyzed by XRD. Due to the small cross-sectional area of the sample, the conventional X-ray diffractometer was not effective. In this study, a micro-area X-ray diffractometer was used to analyze the material phase, and the detection spot was controlled at 20 μm, which can achieve fixed-point measurement.

## 3. Results and Discussion

### 3.1. Microstructure Characteristics of Ti6Al4V/Inconel 718

#### 3.1.1. XRD Phase Analysis

The XRD diffractograms of the sample printed by pulsed laser are shown in Figure 3. It is clear that different peaks were obtained by positioning different points for XRD scanning. Only α and β phases were detected at the top position of the sample, both of which are typical of Ti6Al4V. Two analyses can explain why the secondary phase was not detected by XRD: (1) There were fewer Ni elements in the laser deposition process, while some Ni elements were pulsed and ablated, and the conditions for secondary phase generation could not be reached. (2) Although secondary phase particles were generated during the deposition process, there were so few secondary phases that their presence could not be detected by XRD. The internal microstructure composition of the sample was diversified with the addition of Ni element in Inconel 718 alloy as the probe tip was turned to the middle region of the sample. The two curves in the middle of Figure 3 show, the transition region of the pulsed laser deposited single-layer and Double-layer samples, where the newly generated secondary phases Ti_2_Ni, Ni_3_Ti, and TiNi can be seen by XRD inspection. The η phase, as a eutectic phase, has the chemical formula Ni_3_(Ti,Nb) mainly. Further observation shows that, with the addition of Ni the peak, where the β phase is located tends to move to the right, while the α peak is not significantly affected. Through the study of non-classical phase transformation of titanium alloys by Xiao et al. [18]. We know that Ni, Nb, and Ta are stabilizers of β-phase in titanium alloys. β-phase as a body-centered cubic (BCC) structure, Ni elements will be preferentially enriched around it. At the same time, the Ni (0.162 nm) atomic radius is smaller than the Ti (0.2 nm) atomic radius, resulting in a smaller lattice constant in the β-phase. This well explains the principle that the β-peak is shifted to the right. The α-phase is mainly more sensitive to elements such as O, La, and Al. Additionally, Ni elements are rarely solid soluble in the α-phase, so the lattice constant changes are not obvious. Therefore, the diffraction peaks of α-phase remain unchanged. The diffraction peak at the bottom is the characteristic curve of Inconel 718, without the generation of secondary phases. The γ-phase, as the matrix of the nickel-based alloy, is a face-centered cubic structure. It incorporates a large amount of alloying elements such as Co and Cr. The Laves phase is the typical phase organization of nickel-based alloys in additive manufacturing, which is mainly influenced by the solidification rate. The phase transition of the sample as a whole can be summarized from top to bottom as follows: α + β → α + β + Ti_2_Ni + TiNi + Ni_3_Ti + η → γ + Laves.

Laves phase as a prone to liquefaction cracking causing failure of the mechanical properties of the workpiece, is to be avoided as much as possible. It is usually eliminated by solid solution in heat treatment. As the main phase organization in Ti6Al4V, the morphological structure and content percentage of α-phase and β-phase greatly affect the mechanical properties of its components. By changing the morphology and content of these two phases through suitable heat treatment, the mechanical properties can be significantly improved. Marlo et al. [19]. changed the ratio of the two phases through reheat treatment and found that the β-phase, which is rich in Al and V elements, can significantly improve the hardness.

#### 3.1.2. Optical Microscope Analysis of Microstructure Morphology

The morphological microstructure after pulsed laser deposition is shown in Figure 4. Figure 4 is the result of the use of an optical microscope (Nagoya, Japan) to characterize the macroscopic morphology of the single-layer and double-layer samples.

At the early stage of deposition, the temperature of the nickel alloy substrate (Inconel 718) is low compared to the heat input of the pulsed laser, so the crystal microstructure at the bottom undergoes directional growth by heat conduction. The macroscopic of the sample cross-section is shown in Figure 4, where the sample appears semicircular. Two of the red lines are used as markers for different regions. As shown in Figure 4G–I, the bottom region of the sample is organized with columnar crystals as the main morphology. Some black massive phases are interspersed between the dendritic axes.

Figure 4D–F characterize the intermediate region of the deposited layer, also referred to as the transition zone. This region is where a large amount of Ti6Al4V and Inconel 718 alloys are mixed and a large amount of secondary phases precipitate in this region. The histomorphology of the transition zone shows a cross shape with longer dendritic axes perpendicular to the printing direction. The dendritic axes in the other three directions are shorter and of similar length. The morphology of the microstructure in the transition region is mainly influenced by heat conduction. As a whole, the heat in the transition region can be quickly transferred to the bottom region by heat conduction because of the faster heat dissipation in the bottom region. This results in a large temperature gradient difference in the vertical direction and a smaller temperature gradient in the other directions. The dendritic crystals grow into a cross-shaped phase under the influence of the above factors.

In the upper region of the deposited layer in Figure 4A–C, the continuous flow of argon gas greatly accelerates the cooling rate. At the same time, the heat conduction from the top to the substrate is not obvious due to the obstruction of the transition zone, and the temperature gradient is similar in all directions, resulting in the upper region being dominated by equiaxed crystals.

Figure 5 shows the characterization of the pulsed laser printed Double-layer sample. From Figure 5B, it can be seen that the macroscopic crack in the red circle goes through the cross-shaped phase and extends all the way. Figure 5D,E show the high magnification morphology of the crack surface and the elemental content of the precipitated phase. Cracks can be seen extending along the precipitated phase and part of the precipitated phase is cut through. EDS spectra were performed for a, b, and c in Figure 5D, and the elemental distribution was plotted as Figure 5E. Point a scans the content of elements in the matrix. Point b selects the white blocky phase cut by the crack for scanning. Point c is an element analysis of the black bulk phase. The Cr and Nb of the precipitated phase at point b is higher than that of the deposited layer at point a, which can be considered as enriched Cr and Nb phases. The C element of the precipitated phase at point c is much higher than that of the deposited layer at point a. This part of the phase microstructure can be considered as MC carbide. The above cracks are identified as cold cracks, which are characterized by the formation and expansion around Cr-rich and Nb-rich phases, and they are one of the reasons for the mechanical failure of the workpiece [20]. MC carbides are an important part of nickel alloys. Small amounts of MC carbides provide strengthening by preventing dislocation movement and stabilizing grain boundaries. However, excessive MC content can cause stress concentration and reduce the alloy’s lasting performance. In general, the sediment composition of the sample is the same as expected, and the elemental fluctuations in these parts are due to solidification segregation.

In Figure 5, no oxide or oxygen-rich phases were found inside the microstructure when the double-layer samples were characterized. This indicates that the oxides on the surface of Figure 2 sample are not entrapped within the microstructure. The denseness of the wire material as a raw material is higher than that of the powder material, and the possibility of the wire entering the melt pool with water vapor can be ignored. Then, the formation of this oxide layer is most likely due to the oxidation of the sample by the air in the chamber during natural cooling.

#### 3.1.3. SEM Observation of Phase Transition

Scanning electron microscopy in Figure 6 and Figure 7 demonstrates the histomorphology of the sample printed in a single-layer. Unlike the conventional continuous laser, the pulsed laser has less heat input during the deposition process. The resulting printed samples differ from those observed by other researchers using continuous laser deposition, and the samples from this experiment do not show discrete alpha phases [21]. The elemental detection of the marked parts in Figure 6A,B was performed using an energy spectrometer and then counted in Table 5. The elemental content of Ti in the region indicated by a in Figure 6A reached 97.44 wt%. Additionally, the Ni content in the region indicated by b has increased occupying 22.61 wt%. Wu et al. [21] also made a similar finding that the intracrystalline Ni content is higher than the intergranular Ni content. According to the joint analysis of Table 5 and XRD figure, the elemental content of Ti at point c is 46.68 wt% and the elemental content of Ni is 25.92 wt%, and the phase microstructure at point c is Ti_2_Ni.

Figure 7 shows the SEM and EDS characterization work performed on the sample diffusion region. We selected the cross-like phase and its surrounding area for elemental analysis. With increasing multiplicity, a patterned eutectic phase appeared around the cross-like phase that was not previously observed under the light microscope.

The microstructure is mainly composed of γ isometric crystals and η (Ni_3_Ti) phase precipitated near the γ phase. In addition, a relatively fine irregular phase microstructure was also found. The Cr and Mo contents of the circular phases were significantly higher than the average composition of the matrix material by EDS analysis. The eutectic phases of Cr and Mo are hard and brittle, which have great influence on the mechanical properties of the alloy. In the solidification process, with the formation of Cr- and Mo-rich phases, the edge of the solidification tip evolves into a Cr and Mo depleted zone, and elements such as Ni and Ti increase. Many flower-like η + TiNi eutectic phases were produced under the condition of satisfying the eutectic precipitation formation. Many daisy petal-like η + TiNi eutectic phases were produced under the condition of satisfying the eutectic precipitation formation.

Figure 8 shows the SEM and EDS characterization of the bottom region of the single-layer sample. By the principle of heat conduction, columnar crystals were grown in the bottom region of the sample, and the average width of the columnar crystals was 23.71μm as measured by ImageJ software. Columnar crystals are typical of the microstructure of laser additive manufacturing of nickel-based alloys, which have significant anisotropy. In Figure 8, the light white part (point b) is Laves phase, which is a common phase in the construction of nickel alloy in additive manufacturing technology. They are mainly distributed in the interdendritic region. The black columnar crystal part is the γ phase. According to the element analysis of the marker (a, b) in Figure 8, from the EDS point scan a and b, it can be seen that the enrichment of Nb and Mo in the Laves phase is very high. large mass elements such as Nb and Mo K < 1 diffuse into the interdendritic region during solidification and co-solidify with the remaining metallic liquid. The enrichment of these elements in the dendrites causes a decrease in the melting point and deteriorates the mechanical properties of the component, which requires heat treatment to allow these elements to diffuse fully into the matrix.

### 3.2. Effect of Compositional Changes on the Microhardness of Deposited Layers

Prior to the hardness test, the single-layer samples were scanned using an X-ray energy spectrometer and observed for fluctuations in the major elements of the period.

As can be seen in Figure 9, the Ni element content shows a gradual increase as the scanned sample goes from high to low, while the Ti element shows exactly the opposite fluctuating behavior. Among them, the elemental content curves at the junction of different regions fluctuate greatly. This fluctuation characteristic is most obvious in the bottom region and the middle transition layer region. The upper part as the equiaxed crystal region has a more uniform distribution of each element, and the fluctuation amplitude is not large. The middle region, the region where two different alloy materials fuse and diffuse with each other, generates the secondary phase with a more frequent elemental bias, resulting in larger elemental fluctuations. Similarly, at the bottom, i.e., the columnar crystal region, the interdendritic region produces similar fluctuations as the transition region due to the enrichment of Nb and Mo elements. The elemental content of Cr in the dendritic axis and between dendrites is not very different, so the curve is relatively smooth. In conclusion, the overall distribution of elements is the same as that characterized in the previous sections, so the phase composition at the corresponding positions can be predicted well when performing statistics on hardness.

The hardness of the pulsed laser fusion deposited 50% Ti6Al4V + 50% Inconel 718 (same deposition volume) is shown in Figure 10, with a maximum hardness of 951.4 HV. The hardness does not change significantly in the γ+Laves substrate region, and the slight curve fluctuation may be due to the difference in hardness brought about by the measured columnar and interdendritic regions, with the lowest value of 701.4 HV measured. As the transition layer is approached, the Ti-led elements react with the nickel alloy in brittle phases such as TiNi, Ti_2_Ni, and other secondary phases precipitated, leading to an increase in hardness. When moving to the top layer, the secondary phase precipitation decreases as Ti6Al4V occupies a larger portion, and the equiaxed crystals are distributed as the main microstructure at the top of the sample. The slope of the hardness curve is smaller on the Ti6Al4V side of the microstructure, and the overall hardness is somewhat higher than on the Inconel 718 side. The appearance of high hardness on the Ti6Al4V side and low hardness on the Inconel 718 side is inconsistent with the study by Huang et al. [22]. The authors attribute this occurrence to the fact that the microstructure on the Ti6Al4V side has the same temperature gradient around it during solidification and can cool rapidly under the protective gas, forming snowflake isometric crystals, as shown in Figure 4. When the microstructure of the equiaxed crystal part was measured in a hardness tester, a high level of hardness was obtained.

## 4. Conclusions

In summary, a pulsed laser fusion deposition of Ti6Al4V/Inconel 718 as a functional gradient material was investigated in this paper, and the two wires were fused to a nickel alloy substrate in the same volume. Based on the results obtained, the following conclusions can be drawn:A new pulsed laser melting deposition technique was investigated. Based on this, Ti6Al4V and Inconel 718 with equal volume ratios in gradient materials were successfully printed and investigated.Cold cracking in the transition region due to Nb and Cr element bias enrichment during double-layer sample printing.The sample can be divided into three regions. The bottom region is Inconel 718 with columnar crystals as the main structure. The middle transition region is rich in secondary phases TiNi, Ti_2_Ni, etc., with cross and patterned morphology. The top of the sample has equiaxed crystals as the main microstructure.According to the hardness data, the maximum hardness of 951.4 HV was reached in the transition layer. The top part is second only to the transition zone in hardness due to the presence of equiaxed crystals. The bottom has the softest hardness with the lowest value of 701.4 HV.

## Figures and Tables

**Figure 1 materials-16-02437-f001:**
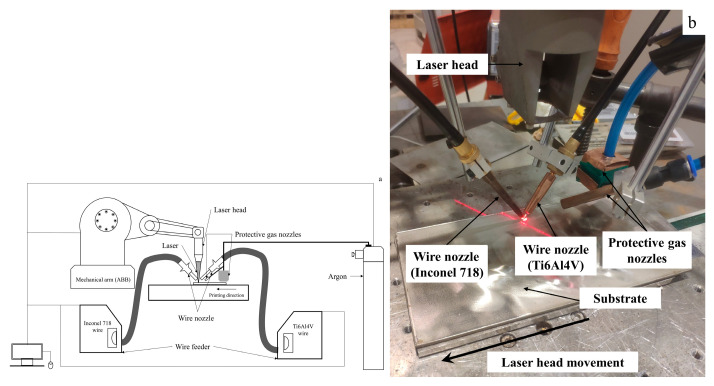
A picture of the experimental set-up (**a**) Schematic diagram of platform equipment; (**b**) Laser equipment operation diagram.

**Figure 2 materials-16-02437-f002:**
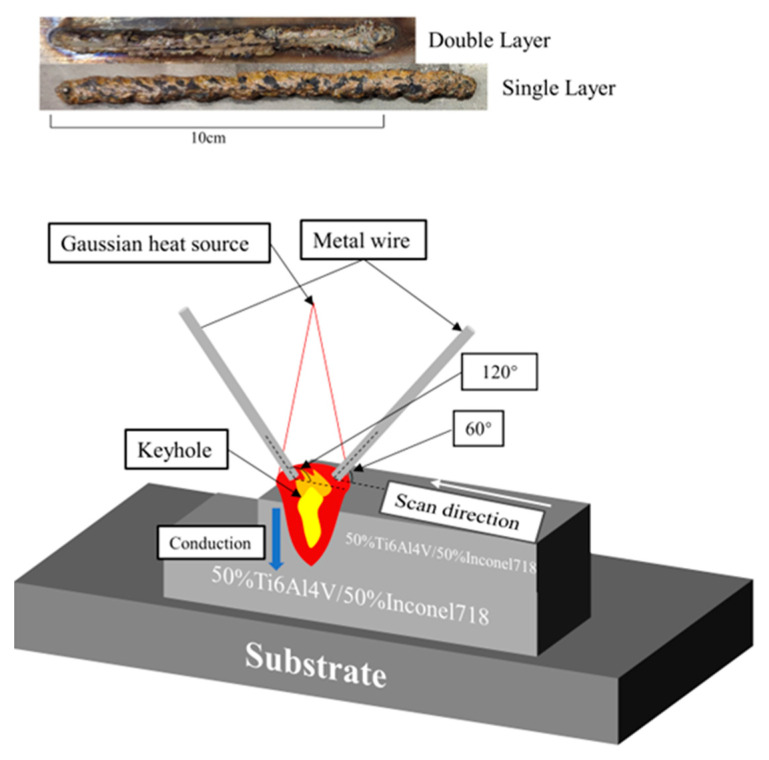
Sample diagram and print process diagram.

**Figure 3 materials-16-02437-f003:**
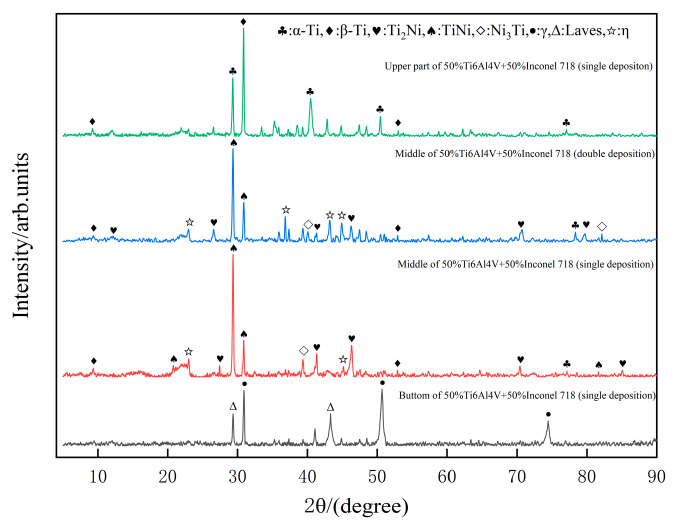
XRD diffraction pattern of single-layer samples.

**Figure 4 materials-16-02437-f004:**
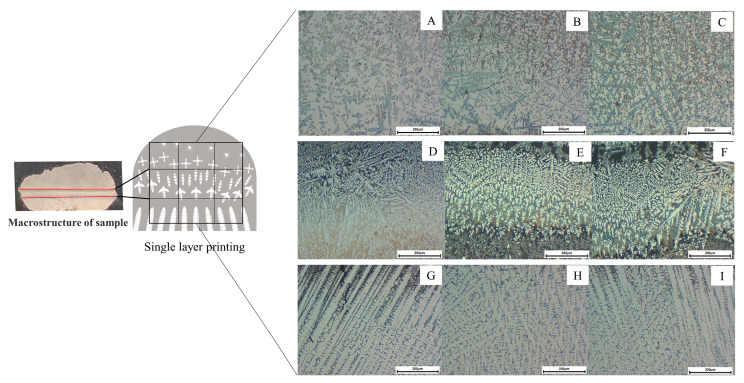
Optical microscopy characterization of single-layer samples. (**A**–**C**) Top equiaxed crystal zone. (**D**–**F**) Middle secondary phase mixing zone. (**G**–**I**) Bottom columnar crystal zone.

**Figure 5 materials-16-02437-f005:**
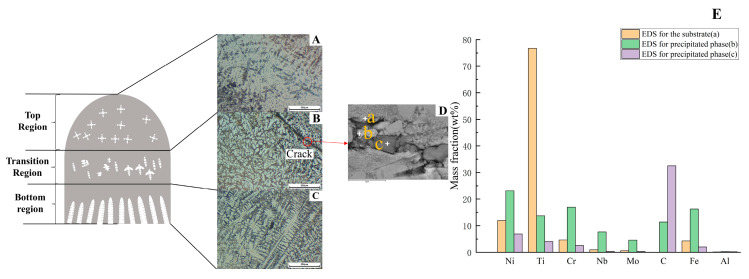
Double-layer sample characterization chart. (**A**–**C**) Optical microscope characterization. (**D**) EDS measurement elements. (**E**) Elemental content comparison chart (point a, b, c).

**Figure 6 materials-16-02437-f006:**
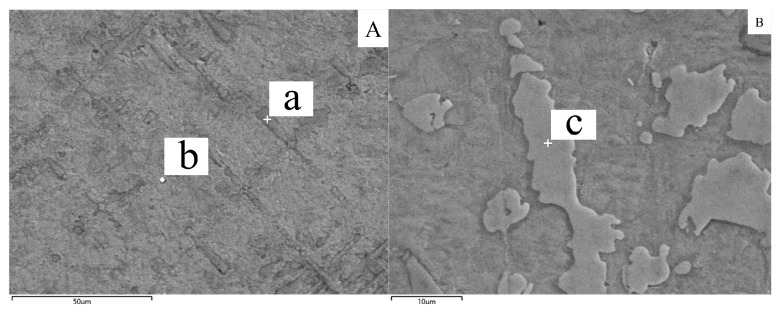
EDS scan single-layer sample element composition. (**A**) Top equiaxed crystal, a: branches of equiaxed crystals. b: Matrix of the sample. (**B**) Precipitated phase at the top of the sample. c: Mark points for measuring precipitated phase elements.

**Figure 7 materials-16-02437-f007:**
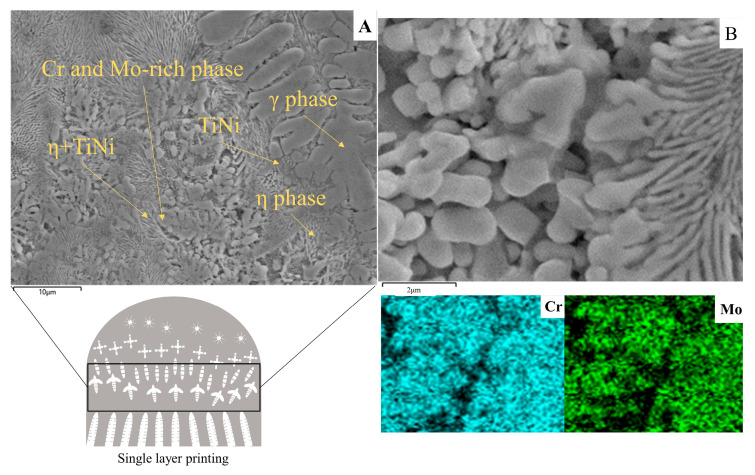
SEM of the central region (**A**) Phase morphology diagram. (**B**) Local enlargement of phase microstructure.

**Figure 8 materials-16-02437-f008:**
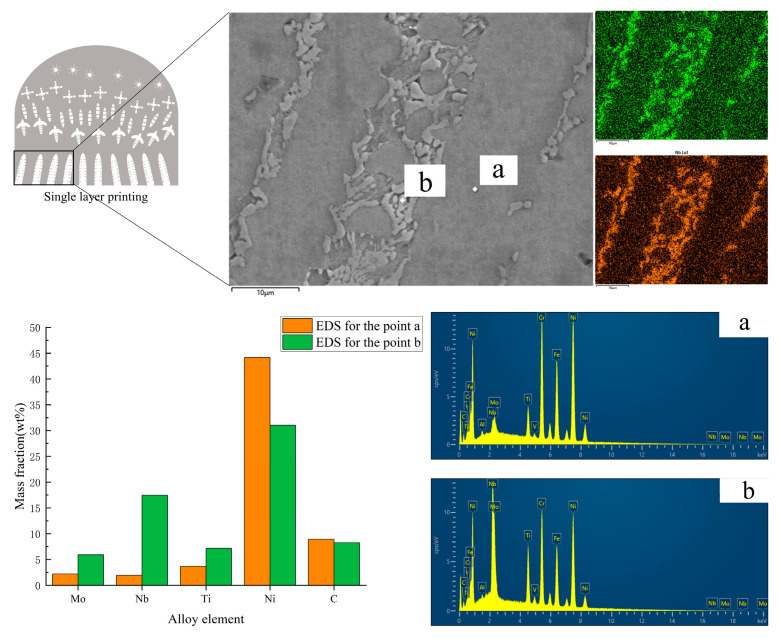
SEM and EDS of the bottom columnar crystal region. a—Columnar crystal branch; b—Interdendritic precipitated phase.

**Figure 9 materials-16-02437-f009:**
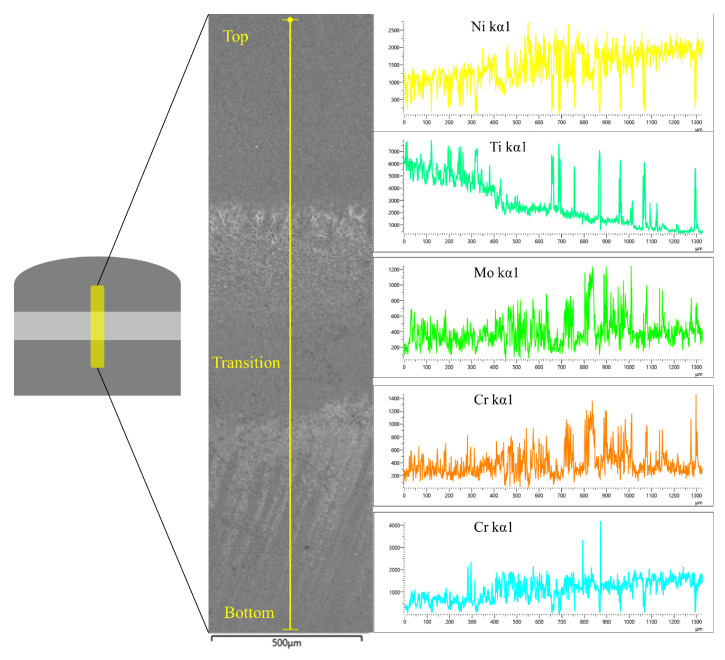
Single-layer sample overall element distribution.

**Figure 10 materials-16-02437-f010:**
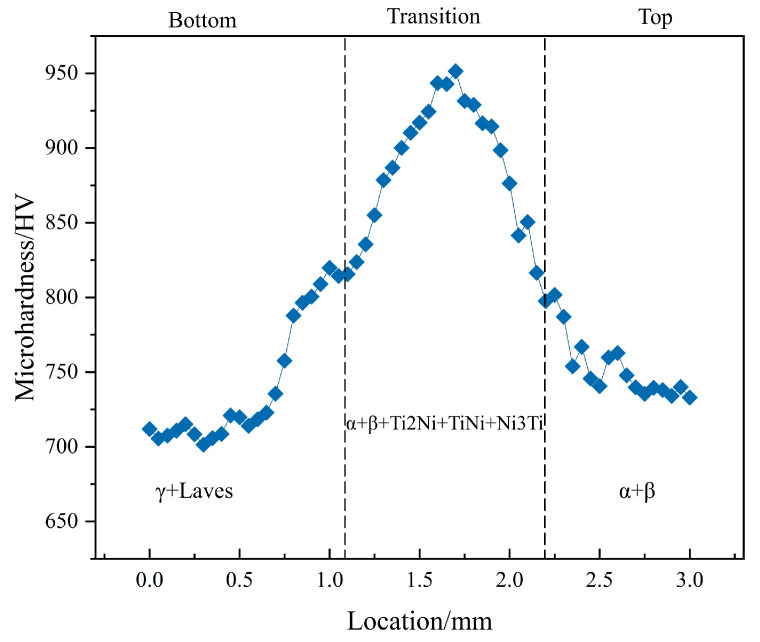
Hardness distribution of Ti6Al4V/Inconel 718 coating.

**Table 1 materials-16-02437-t001:** Chemical composition of the Ti6Al4V alloy wire (mass fraction, %).

Al	V	Fe	C	N	H	O	Ti
5.9	3.97	0.05	0.10	0.05	0.01	0.10	Bal.

**Table 2 materials-16-02437-t002:** Chemical composition of the Inconel 718 alloy wire (mass fraction, %).

Ni	Cr	Nb	Mo	Ti	Al	Si	Fe
53.61	18.76	4.86	3.12	1.05	0.49	0.18	Bal.

**Table 3 materials-16-02437-t003:** Processing parameters.

Laser Power	Laser on Time	Laser Frequency	Arm’s Speed	Wire Feed	Wire Angle	Argon Gas Flow	Amount of Defocusing
480 W	8 ms	20 Hz	2.5 mm/s	10 mm/s	60~120°	20/L·min^−1^	+4 mm

**Table 4 materials-16-02437-t004:** Main physical properties of Ti and Ni [16].

Material	Melting Point/°C	Coefficient of Linear Expansion/ (10^−6^·K^−1^)	Thermal Conductivity/ (W·m·K^−1^)	Specific Heat Capacity/ (J·kg^−1^·K^−1^)	Density/ (g·cm^−3^)
Ti	1677	8.2	13.8	539.1	4.5
Ni	1453	15.3	88	440.0	8.9

**Table 5 materials-16-02437-t005:** Element content at the mark (wt%).

Symbol	Ti	Ni	Cr	Mo	Nb
a	97.44	0.80	0.28	0.09	0.45
b	60.07	22.61	4.91	0.33	1.16
c	46.68	25.92	9.29	2.67	0.89

## Data Availability

Not applicable.

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
