# Peer review of "Study of Phase Evolution Behavior of Ti6Al4V/Inconel 718 by Pulsed Laser Melting Deposition"

_materials, 2023, doi:10.3390/ma16062437_

Round 1
Reviewer 1 Report
The submitted manuscript possesses the aim to elaborate on additive manufacturing technics of the functional gradient material using pulsed laser melting of Ti6Al4V/Inconel 718 alloy wire. The additive manufacturing of the functional gradient material is the relevant topic in the field of materials science and technology. It is worth noting that the additive manufacturing of such composites has been studied earlier in following papers:
- Wu D, Yuan S, Chao Y, et al. Microstructure Evaluation and Mechanical Properties of Ti6Al4V/Inconel 718 Composites Prepared 380 by Direct Laser Deposition[J]. Rare Metal Materials and Engineering, 2021, 50(1): 0078-0084. 381;
- Huang Jinxin, Sun Zhonggang, Chang Hui, et al. Compositional Changes and Microstructure Evolution of Ti6Al4V-Inconel 718 382.Functionally Graded Materials by Laser Additive Manufacturing[J]. Rare Metal Materials and Engineering, 2020, 49(8).
However, wire-feed additive manufacturing technology has been applied for such composites at the first time. The article should be improved before publication in the journal according to the comments:
1. Detailed chemical composition of raw materials (Ti6Al4V and Inconel 718 alloy wires) should be presented in a table.
2. Gas shielding features were described very poor. Gas consumption rate and gas purity should be presented in detail.
3. The yellow oxidation layer on the surface of the wall was observed in Figure 2. The effect should be discussed in the text.
4. The strategy of the additive manufacturing should be described more particularly. The trajectory of laser head, mechanisms of fusion, and how the authors took into consideration difference of thermophysical properties of Ti6Al4V and Inconel 718 alloys were not clarified in the text.
5. Why did the authors apply the certain processing parameters (Table 1)? It should be also defined.
6. Only two layers were built up during applied additive manufacturing process. Is it possible to build up more layers using suggested processing modes?
7. English language and style are fine/minor spell check required.
8. There are some typographical errors in the text.
Reviewer 2 Report
This is a well-written and informative sentence that summarizes the key findings of the study. However, it could be improved by breaking it down into smaller, more digestible chunks to make it easier for readers to follow the flow of the argument.
Here are some suggested revisions:
Start with an introductory sentence that provides context for the study, such as "This study investigates the use of pulsed laser melting deposition for additive manufacturing of Ti6Al4V/Inconel 718 alloy wire."
Divide the description of the sample into separate sentences to highlight the key features of each region, such as "The samples can be divided into three regions based on tissue characteristics after formation. In the bottom region, the alloy grows into columnar crystals by heat conduction. In the middle transition region, the two alloys are mixed to produce many secondary phases, and the tissue shows a cross-like phase. In the top region, the phase organization shows equiaxed crystals."
Explain the significance of the phase organization by providing some context, such as "The phase organization changes from γ+Laves→α+β+Ti2Ni+TiNi+Ni3Ti→α+β from the bottom to the top of the sample, which has important implications for the mechanical properties of the material."
Summarize the key findings about hardness in a separate sentence, such as "The hardness data shows a maximum value of 951.4 HV in the transition zone, while the top part is second to the transition zone in hardness due to the high number of equiaxed crystals. The bottom region is dominated by columnar crystals and is the softest of the three regions, with a minimum hardness value of 701.4 HV."
Consider defining technical terms such as "columnar crystals" and "equiaxed crystals" for readers who may not be familiar with them.

Reviewer 3 Report
The present work entitled "Study of phase evolution behavior of Ti6Al4V/Inconel718 by pulsed laser melting deposition" considers the study of the structure and composition of Ti6Al4V/Inconel 718 coating. Detailed analysis of the structure and phase composition of the resulting coatings has been carried out.
The manuscript is generally well written, with a rather informative introduction. The methodological part is described in detail. The conclusions are consistent with the evidence and arguments presented. I think this paper can be considered for publication, but it needs some improvements.
General remarks:
1. The manuscript requires editing of English language and style. There are many typos.
2. The abstract should be rewritten in order to summarize each component of the paper. An example of the abstract layout can be:
* Short introduction of the activity shown in the paper; * Highlight the aim of the work as the possible final application of the treatment analyzed;
* Brief introduction of the experimental procedures;
* Summary of the results;
3. It is not entirely clear why the authors, when describing the structure of the deposited layer, use the term "tissue".
4. It is not clear why and on what grounds the authors state that the composition of the deposited layer is 50% Ti6Al4V + 50% Inconel 718. This is not entirely correct.
5. There is no analysis of the macrostructure of the deposited layer in the work.
6. What is the point of creating a single-layer and two-layer composite layers.
7. The proposed technology looks rather labor-intensive for obtaining layers on a large surface. Justification needed.
8. Authors use different terms "single-layer" and "mono-layer". Although they are talking about the same.
9. There is no description of the instruments used in the experiment. How X-ray analysis was carried out in various zones of the deposited layer?
10. It is not indicated which wire was fed before and after the laser beam. Does it matter which wire will be before and after the laser beam?
11. Unfortunately, the paper does not discuss thermal stresses in the composite. As well as there is no information about the coefficients of thermal expansion. This will contribute to obtaining strong conclusions and highlights of work.
12. Incorrect table 2 name. Incorrect title of Figure 9. I believe that it can be combined with Figure 8.
13. Line 64: Extra letter T.
14. Line 104: Here the frequency is 12 Hz, and in table 1 - 20 Hz.
15. Table 1: It would be better to specify the speeds in the same units (mm/s).
16. Line 122: Why is this technological parameter (246 mm) not listed in Table 1?
17. Line 135: Here and below, indices in chemical formulas must be subscript.
18. Line 158: Need a capital letter
19. Section 3.1.1: It is not deciphered which phases correspond to the Greek letters and the Laves phase (solid solutions, chemical compounds). The Laves phases have not been characterized in any way.
20. Figure 3: The diffractograms have not been fully deciphered. There are a large number of peaks that do not correspond to any phases.
21. Figure 5,6,7,8,9: Difficult to read inscriptions in the figure.
22 Line 182, 222: Here and below, histomorphology is an incorrect term. (better “microstructure”)
23. Line 119, 126. Text is repeated
The manuscript with opportune modifications can be considered for publication in Materials.
